# Influence of a Commercial Synbiotic Administered In Ovo and In-Water on Broiler Chicken Performance and Meat Quality

**DOI:** 10.3390/foods12132470

**Published:** 2023-06-23

**Authors:** Siria Tavaniello, Davide De Marzo, Marek Bednarczyk, Marisa Palazzo, Sanije Zejnelhoxha, Mengjun Wu, Meng Peng, Katarzyna Stadnicka, Giuseppe Maiorano

**Affiliations:** 1Department of Agricultural, Environmental and Food Sciences, University of Molise, 86100 Campobasso, Italy; siria.tavaniello@unimol.it (S.T.); m.palazzo@unimol.it (M.P.); s.zejnelhoxha@studenti.unimol.it (S.Z.); m.wu@studenti.unimol.it (M.W.); m.peng@studenti.unimol.it (M.P.); 2Department of Precision and Regenerative Medicine and Jonian Area, Section of Veterinary Science and Animal Production, University of Bari ‘Aldo Moro’, 70010 Valenzano, Italy; davide.demarzo@uniba.it; 3Department of Animal Biotechnology and Genetics, Bydgoszcz University of Science and Technology, 85-084 Bydgoszcz, Poland; marbed13@op.pl; 4Department of Oncology, Faculty of Health Sciences, Collegium Medicum Nicolaus Copernicus University, Łukasiewicza 1, 85-821 Bydgoszcz, Poland; katarzyna.stadnicka@cm.umk.pl

**Keywords:** broiler chickens, in ovo synbiotic, hatchability, growth performance, meat quality

## Abstract

The present study aimed to test the synbiotic PoultryStar^®^ sol^US^ delivered in ovo to evaluate its effect on hatchability, productive performance and meat quality, compared to its post-hatch administration in water. On the twelfth day of embryonic incubation, 1200 fertile eggs were divided into synbiotic groups injected with 2 mg/embryo (T1) and 3 mg/embryo (T2), a saline group injected with physiological saline and an uninjected control group (C). After hatching, 120 male chicks/group were reared and chicks from the saline group were supplemented with the synbiotic via drinking water (T3). Hatchability was low in both T1 and T2 groups. Growth performance was not affected by the treatments. However, in the second rearing phase (15–36 days), birds from the C and T3 groups were heavier than T1 birds, due to a higher feed intake and daily weight gain. Neither route of synbiotic administration influenced final body weight (at 56 days), weight and yield of the carcass or commercial cuts. Physico-chemical properties, total lipid, cholesterol and fatty acid composition of breast muscle were not affected by the treatments. Considering its exploratory nature, this study has raised many questions that need further investigation, such as the bioactive combination and the effect on embryonic development.

## 1. Introduction

In the last decade, to tackle antimicrobial resistance, considerable scientific progress has been made towards finding non-antibiotic alternatives as approaches to enhance animal immunity and increase disease resistance, while improving production. Among many different non-antibiotic alternatives, there is a great potential for probiotics, prebiotics and synbiotics in poultry production [1]. The current scientific definition of a synbiotic is a mixture comprising live microorganisms and substrate(s) selectively utilized by host microorganisms that confers a health benefit on the host [2]. There are several ways to deliver synbiotics into the avian gastrointestinal tract. In chickens, they are administered routinely in feed or in water immediately after hatching. However, there is a growing interest in delivering these microflora-promoting bioactives as early as possible, during the embryonic stage, to favor the establishment of a beneficial microflora capable of counteracting possible pathogens after hatch [3]. The in ovo technology, developed by our group, is based on a single dose of prebiotics or synbiotics injected on the twelfth day of embryonic development into the egg air cell [4,5,6,7]. This technology offers advantages on several levels as compared to post-hatch administration, including uniformity and precision of bioactives delivery to each embryo, low dose of bioactive compound, and optimal timing for stimulating gut colonization with beneficial bacteria [6]. The research on in ovo stimulation and the lifelong effects support this method as efficient programming of lifespan conditions in commercially raised chickens. In ovo administration of bioactive substances can result in improvements in hatchability [8], hatching weight [9], growth traits and feed efficiency [5,10], intestinal morphology [9], muscle microstructure, microvascularization and meat quality [11,12,13,14,15,16], immune system development [17,18], physiological characteristics [19,20], and transcriptome modulation [21,22], as well as proteome changes [23] of broiler chickens. Moreover, an optimally functioning immune system will affect the immune competence of one-day old chicks and thereby their ability to respond to vaccination and infections. The aforementioned data were collected under fully controlled conditions. Few studies have dealt with the effects of commercially available synbiotics, injected in ovo, on the productive performance and meat quality traits of broiler chickens reared under semi-intensive systems. Previous studies suggested that the commercially available synbiotic product PoultryStar^®^, (BIOMIN GmbH, Herzogenburg, Austria), administered in water or in feed in broiler chickens [24,25,26,27] could be a natural alternative to antibiotic growth promoters. This synbiotic is a well-defined, poultry-specific, multi-species product. The soluble form of it (PoultryStar^®^ sol^US^, BIOMIN GmbH, Herzogenburg, Austria) is recommended to be used in drinking water. The objective of this study was to compare two different routes of administration (in ovo vs. the recommended in water administration) using two doses for the in ovo injection of the commercial synbiotic PoultryStar^®^ sol, documenting the effects on hatchability, growth performance and meat quality traits of broiler chickens reared under semi-intensive conditions.

## 2. Materials and Methods

### 2.1. Ethics

The experimental procedures were approved by the ethical commission of the University of Molise (prot. n. 31905, 2 September 2021) and in accordance with the European Commission guidelines (2010/63/EU).

### 2.2. Birds and Experimental Design

Fertilized eggs (average weight: 68 g) obtained from the same breeder flock (Ross 308) were incubated in standard conditions (37.5 °C, 55% relative humidity, turned every hour, for 18 d, then in the hatcher for 3 d at 36.9 °C, 65% relative humidity). On the twelfth day of embryonic development, after candling, 1200 eggs with viable embryos were randomly divided into the following 4 experimental groups: synbiotic group (T1) injected with a single dose of 2 mg/embryo suspended in 0.2 mL of physiological saline; synbiotic group (T2) injected with a single dose of 3 mg/embryo suspended in 0.2 mL of physiological saline; saline group injected with 0.2 mL of physiological saline (0.9% NaCl); control group (C), uninjected. Saline and synbiotic solutions were injected into the air chamber, and the hole was sealed with organic glue. The injection dose used was based on previous research carried out with different synbiotics [7,11,14]. The synbiotic preparation administered in ovo and in water (PoultryStar^®^ sol^US^, BIOMIN GmbH, Herzogenburg, Austria) included a prebiotic (2 × 10^11^ CFU kg/g of fructooligosaccharides, FOS) and a probiotic mixture of 4 microbial strains selected from 4 different sections of the poultry gastrointestinal tract: *Pediococcus acidilactici* isolated from the cecum, *Bifidobacterium animalis* from the ileum, *Enterococcus faecium* from the jejunum and *Lactobacillus reuteri* from the crop. Hatchability was calculated as the proportion of hatched chicks to the number of fertile eggs, candled at 12 days of incubation. After hatching, all the chicks were sexed and vaccinated according to current commercial practice. A total of 480 male broiler chickens were randomly chosen, 120 from each experimental group (T1, T2, C, saline). After hatching, chicks from saline group, used as a positive control for the determination of hatchability, were supplemented with the synbiotic via drinking water (group T3). According to the manufacturer’s recommendations, the schedule application for the T3 group was, in newly hatched chicks, for the first 3 d of life and around feed changes for 3 consecutive days, starting 1 d before the feed change. For the intermittent application, the recommended dose is 20 g/1000 birds/day. Chicks were reared under semi-intensive rearing conditions in floor pens (10 replications for each experimental group of 12 chicks) with 2 open sides; temperature, humidity and lighting were not controlled. The trial was carried out in a private poultry farm (700 m altitude) located in the Molise region (Southern Italy), from September to November 2021. The feed was a commercial complete mix formulated according to age (Table 1). 

Performance parameters (body weight (BW), feed intake (FI)) were recorded on a pen basis on days 0, 14, 36, and 56 of age. Daily weight gain (DWG), daily feed intake (DFI), and feed conversion rate (FCR) corrected for mortality were calculated. Daily mortality, expressed as percentage, was registered for the whole rearing period.

### 2.3. Slaughter Surveys and Physico-Chemical Analyses

At 56 d of age, a total of 20 randomly chosen birds from each group (2 birds for each replication) were individually weighed (after a fasting period of 12 h), and transported (including careful catching and loading) for 30 km to a commercial poultry slaughterhouse. Birds were electrically stunned and slaughtered. The carcass weight was obtained by removing the head, neck, shanks and abdominal fat from bled, plucked and eviscerated chickens; carcass yield was calculated. Then, the breast (including the pectoralis major and pectoralis minor muscles) and leg (thigh + drumstick) weights were recorded; cut yields were calculated based on hot carcass weight. The right pectoral muscle (PM) was used to record pH and color 24 h post-mortem, according to the method used by Tavaniello et al. [15]. The left PM was vacuum packaged and stored frozen (–20 °C) until chemical analysis.

### 2.4. Total Lipids and Fatty Acid Profile

Lipid extraction from the breast muscle was performed using the chloroform: methanol extraction procedure [28]. Fatty acids (FA) were quantified as methyl esters (FAME) using a gas chromatograph GC Trace 2000 (ThermoQuest EC Instruments, Milan, Italy) equipped with a flame ionization detector (260 °C) and a fused silica capillary column (Zebron ZB-88, Phenomenex, Torrance, CA, USA) of 100 m × 0.25 mm × 0.20 μm film thickness. Helium was used as a carrier gas. The column temperature was held at 100 °C for 5 min, then raised by 4 °C/min up to 240 °C and maintained for 20 min. The individual FA peaks were identified by comparison of retention times with those of FAME authentic standards run under the same operating conditions. Results were expressed as percentages of the total FA identified. To assess the nutritional implications, the ratio of n-6 PUFA to n-3 PUFA (n-6/n-3), the ratio of polyunsaturated fatty acids (PUFA) to saturated fatty acids (SFA) (PUFA/SFA) and the atherogenic (AI) and thrombogenic (TI) indices [29] were calculated.

### 2.5. Statistical Analyses

Data were analyzed by one-way ANOVA. Scheffe’s test was applied to compare the differences among means. For performance parameters, each pen was considered as the experimental unit. For carcass and meat quality traits, each bird was considered as the experimental unit. Data were presented as mean and standard error of the mean (SEM) and differences were considered significant at *p* < 0.05. Statistical analyses were conducted by IBM SPSS Statistic Data Editor version 25 (Chicago, IL, USA).

## 3. Results and Discussion

### 3.1. Hatchability

The groups that were in ovo injected with the commercial synbiotic (T1, T2) showed numerically lower hatchability compared to the control and saline groups (Figure 1). 

Similar results were obtained in a trial conducted in Poland using the same doses of synbiotic (PoultryStar^®^ sol^US^, BIOMIN GmbH, Herzogenburg, Austria) in laying hens (personal communication by Prof. Marek Bednarczyk). This difference between the treated groups and the control groups (positive and negative) could be linked to the nature of the synbiotic used. Triplet et al. [30] found that an in ovo injection on day 18 of incubation with various probiotic bacteria had different effects on hatchability; certain strains of the probiotic negatively impact the hatchability, perhaps as a result of competition for nutrients that are vital for hatch or the secretion of microbial by-products that are toxic for the embryo. The synbiotic used in this study is a well-defined, poultry-specific and multi-species product, including FOS prebiotic and a probiotic mixture of four microbial strains selected from four different sections of the poultry gastrointestinal tract. The delivery route recommended by the manufacturer is post-hatch administration in water; however, since the product is characterized by high solubility, in this study we tested in ovo injection as a perinatal stimulation for optimal gastrointestinal tract development and microbial colonization in early life. The tested doses chosen in this trial are based on the results obtained from our previous studies considering different kinds of synbiotics and prebiotics [6,14,15]. Standardized protocols for in ovo injection of bioactives on day 12 of embryonic development result in high hatchability scores, as in ovo injection does not penetrate the inner parts of the egg, which could potentially disturb the viability of the embryo and decrease hatchability [7].

### 3.2. Growth Performance

Table 2 shows the effects of the synbiotic delivered in ovo at two different doses, and in water, on in vivo performance of broiler chickens in three feeding phases. BW of the newly hatched chicks, ranging from 50.8 g to 51.2 g, was not influenced (*p* > 0.05) by in ovo manipulation. In the first two weeks of life, no effect of synbiotic administration on animals’ growth was found. On the contrary, in the second rearing phase, birds from the C and T3 groups were heavier than T1 birds (*p* < 0.05); this result was the consequence of the higher (*p* < 0.05) DWG and DFI registered for the second feeding period (15–36) in both groups. Final BW was similar among the experimental groups (*p* > 0.05). However, even though we did not find statistical evidence that synbiotic in water improved BW in chickens (*p* = 0.061), there was a numerical improvement in BW on day 56 compared to the control group (+7.3%). FCR was not affected by synbiotic delivered either in ovo or in water (*p* > 0.05).

In general, growth performance recorded in the present trial is slightly lower than that of the standard Aviagen^®^ performance for Ross308 male chickens. This may be related to the semi-intensive rearing system, where environmental conditions were not fully controlled. 

Results from our previous studies using different synbiotics injected in ovo have described contrasting results. Dunislawska et al. [6] found that in ovo injection of synbiotics composed of *Lactobacillus* species combined with galactooligosaccharides (GOS) or raffinooligosaccharides (RFO) did not affect final BWG and FCR of Cobb500 broilers. Maiorano et al. [11] found that final BW of Ross 308 broilers was not influenced by synbiotics injected in ovo (homemade synbiotic: RFO + *Lactococcus lactis* ssp. lactis SL1, RFO + *Lactococcus lactis* ssp. Cremoris IBB SC1; commercial synbiotic: Duolac, containing *Lactococcus acidophilus* + *Streptococcus faecium* with lactose), while RFO + *Lactococcus lactis* ssp. *Cremoris* IBB SC1 and the commercial synbiotic significantly increased the FCR. On the other hand, Stefaniak et al. [31] observed that Ross 308 broilers had lower BW at 35 days of life following a synbiotic injected in ovo composed of 0.528 mg/egg GOS and 1000 CFU/egg *Lactococcus lactis* subsp. *cremoris* IBB SC1 compared to the control, while the BW did not differ in the group that received a synbiotic composed of 1.76 mg/egg inulin and 1000 CFU/egg *Lactococcus lactis* subsp. *lactis* IBB SL1 compared to the control group. In a recent study, Duan et al. [32] found that intra-amniotic in ovo injection (at 18.5 days of embryonic development) of a synbiotic (1 × 10^6^ CFU of *Lactobacillus plantarum*/egg + 2 mg/egg of Astragalus polysaccharide) did not affect the hatching or growth performance of the chicks but significantly increased their FI, BWG, and FCR in the periods 8–14 and 15–21 days post-hatch. Calik et al. [33] did not observe any significant interaction between intra-amniotic administration and dietary synbiotic supplementation (*Enterococcus faecium* + inulin) in broiler performance; however, intra-amniotic synbiotic administration had a positive effect on FCR during days 0–42 and 22–42. Intestinal integrity and cecal microflora were positively influenced by synbiotic treatment. Nevertheless, these results can depend on the combination of prebiotic and probiotic injected in ovo, similarly to the effect of bioactives administered in feed [34]. Regarding the post-hatch administration (in water or in feed) of synbiotics, published studies report conflicting results. Cheng et al. [35] found that broiler chickens (Arbor Acres Plus) fed a diet supplemented with synbiotic (1.5 g/kg synbiotic, consisting of probiotics, *Bacillus subtilis, Bacillus licheniformis* and *Clostridium butyricum*, and prebiotics, yeast cell wall and xylooligosaccharide) showed an increase in average DWG but a reduction of feed/gain ratio from 1 to 42 days of age compared to those fed the baseline diet. A combination of prebiotic (FOS) and probiotic (mixture of four different microorganisms) in broiler diets improved BW, BWG and feed intake [36]. In a study carried out on laying hens, Luoma et al. [37] found that birds fed synbiotics (PoultryStar^®^ me) at 18 and 20 weeks of age showed higher BW in both groups with and without a salmonella challenge as compared to the un-supplemented groups; after 20 weeks of age, BW did not differ significantly among the different treatment groups. In a recent study [27] carried out on three Ross 308 broiler flocks, reared under field conditions, it was found that synbiotic-fed chickens (PoultryStar^®^ sol^US^ in drinking water) performed better in terms of body weight gain, feed conversion ratio and survivability. However, Brugaletta et al. [38] did not find any significant variation in terms of productive performance of Ross 308 broilers between control and synbiotic groups (PoultryStar^®^ HatcheryEU sprayed as gel droplets onto newly hatched chicks; PoultryStar^®^ meEU, supplemented in feed during the entire rearing period); the treatment combination (sprayed + in feed supplementation) improved FCR from 14–29 days, with a positive effect also for the whole rearing period, as compared to the control group. From reviewing published studies and those performed by our group, the variable effects of synbiotics administered in ovo or in feed on broiler performance can be related to different factors such as type and dose of synbiotic, environmental factors and endogenous factors related to animals, but also to the complexity of the interactions that take place in the gastrointestinal tract [6,7,39]. However, we can confirm that the goal of administering the synbiotic in ovo, which was mainly aimed at improving and maintaining animal health status, has been achieved. In fact, the mortality for the whole rearing period, which is one of the basic welfare parameters that indicate the health status of the flock, was slightly lower (*p* = 0.098) in the groups treated with the synbiotic in ovo (5.7%) and in water (5.9%) than in the control group (7.6%). A similar trend (control: 8.5%; synbiotic: 5.8%) was found by Stefaniak et al. [31].

### 3.3. Carcass Traits and Physico-Chemical Properties of Breast Muscle

The post mortem performance of the slaughtered animals is shown in Table 3. The synbiotic treated chickens (T1, T2 and T3) were heavier and had heavier carcasses (+ 7.9%) than those of the control group, however, the differences were not significant (*p* > 0.05). The carcass yield, the weight of the main commercial cuts (breast and legs) and the respective yields, calculated on the carcass weight, did not differ (*p* > 0.05) among groups. In our previous study [14] no effects of in ovo injection of synbiotics were found for carcass and PM weight and yield. However, Maiorano et al. [11], observed a lower carcass yield and a higher PM yield in the group injected in ovo with a commercial synbiotic. As for in feed supplementation of synbiotic, Cheng et al. [35] found a higher breast yield and a lower abdominal fat in broilers supplemented with synbiotic (*Bacillus subtilis, Bacillus licheniformis, Clostridium butyricum* + yeast cell wall and xylooligosaccharide); in addition, no differences in the carcass yield, leg muscle yield, subcutaneous fat thickness or intermuscular fat were found between control and synbiotic groups. Fornazier et al. [40] reported that the inclusion of a synbiotic (*Bacillus subtilis* 4.0 × 10^7^ UFC/g and *Enterococcus faecium* 3.5 × 10^7^ UFC/g; carbohydrase 150 U/g, protease 130 U/g and phytase 0.7 U/g; *Saccharomyces cerevisiae*; mannan oligosaccharides and beta glucans) in the diet (1.5 kg/t) led to improvements in carcass and breast yields of Cobb 500 chickens slaughtered at 42 days old.

The pH and color were not affected by synbiotic administration (Table 4). The ultimate pH (pH_24_) values are within the normal range for poultry meat, with no evidence of preslaughter stress. Likewise, the values observed for color, one of the fundamental indicators in guiding the consumer’s purchasing choices, fall within those indicated in the literature for a meat free from anomalies. According to the literature, the lightness values (L *) of normal chicken meat are between 46 and 53 [41,42]. In our previous studies [11,14], we did not find any significant effect of synbiotic in ovo injection on ultimate pH and color coordinates. Li et al. [43] reported that male Partridge Shank broilers fed a baseline diet supplemented with synbiotic had a higher meat pH_24_ value. 

Table 5 shows the effects of in ovo and in water administration of the same commercial synbiotic formulation on total lipid and cholesterol content, fatty acids composition (% of total fatty acids) and nutritional ratios in PM of broiler chickens. 

Intramuscular cholesterol content, ranging from 59.93 to 63.84 mg/100 g, was not affected (*p* > 0.05) by synbiotic administration. Similarly, in our previous study using different kinds of synbiotics and prebiotics injected in ovo, we did not find any significant effect on cholesterol content of breast muscle in Cobb broiler chickens [14] or Ross broiler chickens [11]. A recent study [44] found that birds fed a diet supplemented with 0.2% mannan-oligosaccharides (MOS) with 10^6^ or 10^7^ CFU *Bifidobacterium bifidum*/g feed showed lower serum triglyceride and total cholesterol levels, improving serum or plasma health indices compared to control birds. As regards the total lipid content, synbiotic treatment had no effect on it (*p* > 0.05). In our previous study [14], we found a lower lipid content in meat from two chicken groups in ovo injected with two different synbiotic formulations, compared to that of the control group. Recently, it has been reported [44] that probiotics can potentially regulate lipid metabolism in chicken by altering the expression of genes involved in lipogenesis, lipolysis, and transport of lipids, which can in turn have hypolipidemic effects and improve the health value of chicken meat. Growing evidence of the gut microbiota–muscle axis offers the possibility of improving the meat quality of broilers through gut microbiota regulation [45]. In terms of human health, the fatty acids composition of meat products is an important parameter of meat quality. Concerning the FA profile (Table 5), total SFA, PUFA and MUFA contents were similar (*p* > 0.05) among treatment groups. On the contrary, in our previous study [14], we found that the group in ovo injected with *Lactobacillus salivarius* + GOS displayed a meat with a higher amount of total SFA and lower amounts of MUFA, PUFA, total n-6 and n-3 PUFA, whereas an in ovo injection with *Lactobacillus plantarum* + RFO did not affect the FA profile of chicken meat, as compared to the control group. Considering the composition of single FA, among SFA, palmitic (C16:0), and stearic (C18:0) acids were the most abundant, while other detected SFA (C14:0, C17:0, C20:0) comprised less than 0.6%. Palmitic acid content, ranging from 23.75% and 25.68%, was not affected by synbiotic administration, whereas, stearic acid content was higher (*p* < 0.05) in the T1 group (10.29%) as compared to T3 (8.27%), with intermediate values for T2 and C (*p* > 0.05). Myristic acid (C14:0) content was higher (*p* < 0.05) in the T2 group as compared to T3. As regards the single MUFA, an increase in the levels of palmitoleic acid (16:1) was observed in the T2 group as compared to the T1 (*p* < 0.01) and T3 (*p* < 0.05) groups. These differences could be due to a different activity of Δ-9 desaturase (stearoyl-CoA desaturase) [46]. No significant differences were found for oleic acid (C18:1), which was the most abundant MUFA, ranging from 30.72% to 33.03%. However, in a recent study [44], it was found that the supplementation of dietary prebiotic MOS along with either 10^6^ or 10^7^ CFU *Bifidobacterium bifidum*/g feed resulted in the upregulation of stearoyl CoA (∆9) desaturase-1 hepatic expression, which catalyzes the biosynthesis of MUFA from corresponding SFA. In fact, an increase in MUFA content was found, at the cost of SFA content. Regarding PUFA, the precursor of the n-6 family (linoleic acid, LA, C18:2) was the most abundant, ranging from 23.31% to 24.46% but was similar among groups (*p* > 0.05), followed by arachidonic acid (C20:4n-6), ranging from 3.56% to 4.26%. Synbiotic treatment affected the levels of C20:2 n-6 (*p* < 0.05) and C22:4 n-6 (*p* = 0.065), even though the differences within groups were not significant. The n-3 family precursor (linolenic acid, ALA, C18:3) and the long-chain n-3 PUFA (C20:5, C22:5 and C22:6) were not affected by synbiotic treatment (*p* > 0.05). Total n-6 and n-3 PUFA contents were also similar among groups. Similarly, Dev et al. [44] did not find any significant effect of synbiotic administration on PUFAs; in addition, the authors detected that the ∆5  +  ∆6 desaturase index of chicken meat, associated with catalysis of PUFAs synthesis, was not affected by synbiotic supplementation. 

The health-related indices (n-6/n-3, P/S, AI and TI) were not influenced by synbiotic administered in ovo or in water. On the other hand, in our previous study [14] we found that the group injected in ovo with *Lactobacillus salivarius* + GOS developed meat with a lower P/S, and higher AI and TI, whereas the other synbiotic preparation (*Lactobacillus plantarum* + RFO) did not affect the FA composition or nutritional profile of chicken meat, as compared to the control group. Dev et al. [44] found that the PUFA:SFA ratio exhibited a progressive increase, while the AI and TI revealed a progressive decrease from treatment T1 (control) to the other experimental groups administered different doses of MOS and probiotic (*Bifidobacterium bifidum*). The PUFA n-6/n-3 ratio was similar among groups and particularly high (15.20–16.38) considering the current nutritional recommendations for human diets, which should not exceed 4.0. The PUFA/SFA values were favorably high (0.87–0.95), as they should be increased to above 0.4 [47]. Furthermore, the low AI and TI values found in our study indicated a better nutritional quality of the meat.

## 4. Conclusions

The multi-species synbiotic PoultryStar^®^ sol (BIOMIN GmbH, Herzogenburg, Austria), tested by in ovo injection, negatively influenced hatchability at both doses tested. Neither administration route of the commercial synbiotic negatively affected productive performance and meat quality. Considering its exploratory nature, this study has produced many questions in need of further investigation, including how this synbiotic acts during the embryonic development, the reasons for reducing hatchability and the optimal dose for the in ovo injection. This would broaden the knowledge of bioactives that can be effectively used for in ovo technology that do not impair hatchability, but do promote intestinal health, immunological function and production performance of chickens, with the aim of reducing the use of antibiotics in poultry production.

## Figures and Tables

**Figure 1 foods-12-02470-f001:**
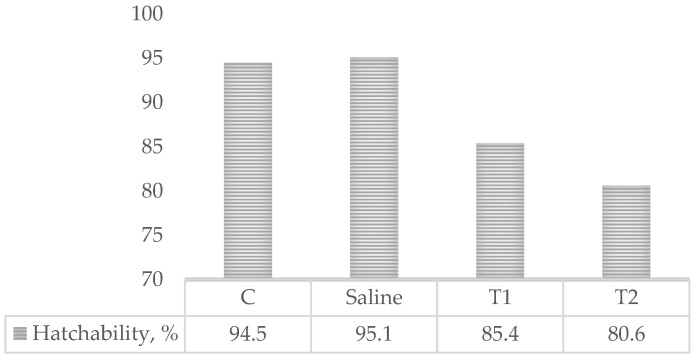
Hatchability rate. C =Control (untreated); Saline = in ovo injected with physiological saline; T1 = in ovo injected with synbiotic (2mg/embryo); T2 = in ovo injected with synbiotic (3mg/embryo).

**Table 1 foods-12-02470-t001:** Diet supplied to the birds.

	Period		
	Starter (0–14 d)	Grower (15–36 d)	Finisher (37–56 d)
Dietary components, kg			
Corn	42.17	34.96	12.73
White corn	0.00	0.00	15.00
Wheat	10.00	20.00	25.01
Sorghum	0.00	0.00	5.00
Soybean meal	23.11	20.63	17.60
Expanded soybean	10.00	10.00	13.00
Sunflower	3.00	3.00	3.00
Corn gluten	4.00	3.00	0.00
Soybean oil	3.08	4.43	5.48
Dicalcium phosphate	1.52	1.20	0.57
Calcium carbonate	0.91	0.65	0.52
Sodium bicarbonate	0.15	0.10	0.15
Salt	0.27	0.27	0.25
Choline chloride	0.10	0.10	0.10
Lysine sulfate	0.59	0.55	0.46
DL-methionine	0.27	0.29	0.30
Threonine	0.15	0.14	0.14
Enzyme-roxazyme G2g	0.08	0.08	0.08
Phytase 0.1%	0.10	0.10	0.10
Vitamin–mineral premix ^a^	0.50	0.50	0.50
Calculated nutrient content			
Dry matter, %	88.57	88.65	88.64
Crude protein, %	22.70	21.49	19.74
Lipid, %	7.06	8.24	9.74
Fiber, %	3.08	3.04	3.07
Ash, %	5.85	5.17	4.49
Lysine, %	1.38	1.29	1.21
Methionine, %	0.67	0.62	0.59
Methionine + cysteine, %	1.03	0.97	0.91
Calcium, %	0.91	0.80	0.59
Phosphate, %	0.63	0.57	0.46
Metabolizable energy, Kcal/kg	3.076	3.168	3.264

^a^ Provided per kg of diet: vitamin A (retinyl acetate), 13,000 IU; vitamin D3 (cholecalciferol), 4000 IU; vitamin E (DL-α_tocopheryl acetate), 80 IU; vitamin K (menadione sodium bisulfite), 3 mg; riboflavin, 6 mg; pantothenic acid, 6 mg; niacin, 20 mg; pyridoxine, 2 mg; folic acid, 0.5 mg; biotin, 0.10 mg; thiamine, 2.5 mg; vitamin B12 20 µg; Mn, 100 mg; Zn, 85 mg; Fe, 30 mg; Cu, 10 mg; I, 1.5 mg; Se, 0.2 mg; ethoxyquin, 100 mg.

**Table 2 foods-12-02470-t002:** Effect of delivery route and dose of synbiotic on growth performance of broiler chickens.

	Group ^a^					
	C	T1	T2	T3	SEM	*p*-Value
Body weight, g	no. 120	no. 120	no. 120	no. 120		
d 0	50.81	51.21	50.97	50.68	0.17	0.799
d 14	467.00	475.41	492.86	489.60	5.89	0.384
d 36	2154.78 ^a^	2002.36 ^b^	2118.70 ^ab^	2206.81 ^a^	25.28	0.004
d 56	4006.03	3977.78	4152.78	4297.22	50.56	0.061
Daily weight gain, g/bird/day						
d 0–14	29.73	30.30	31.56	31.35	0.42	0.390
d 15–36	140.65 ^a^	127.25 ^b^	135.49 ^ab^	143.10 ^a^	2.00	0.002
d 37–56	92.56	98.77	101.70	104.52	2.17	0.131
d 0–56	70.63	70.12	73.25	75.83	0.90	0.060
FCR ^b^						
d 0–14	1.28	1.28	1.30	1.29	0.00	0.481
d 15–36	1.61	1.60	1.59	1.60	0.00	0.454
d 37–56	2.00	1.97	2.00	1.98	0.01	0.301
d 0–56	1.63	1.62	1.63	1.63	0.00	0.297
Daily Feed intake, g/bird/day						
d 0–14	38.15	38.89	40.91	40.46	0.56	0.217
d 15–36	225.79 ^a^	203.78 ^b^	215.21 ^ab^	229.20 ^a^	3.33	0.003
d 37–56	185.03	194.42	203.30	207.51	4.77	0.212
d 0–56	115.05	113.40	119.24	123.31	1.58	0.068

^a^ C = Control (untreated); T1 = in ovo injected with synbiotic (2 mg/embryo); T2 = in ovo injected with synbiotic (3 mg/embryo); T3 = in water synbiotic. ^b^ FCR = feed conversion ratio. SEM = standard error of the mean; ^a,b^ Means within a row lacking a common superscript differ (*p* < 0.05).

**Table 3 foods-12-02470-t003:** Effect of delivery route and dose of synbiotic on carcass traits of broiler chickens.

	Group ^a^					
	C	T1	T2	T3	SEM	*p*-Value
Final body weight, g	3807.99	4108.88	4053.86	4136.70	69.72	0.333
Carcass weight, g	2755.67	2983.67	2939.17	2998.67	49.98	0.294
Carcass yield, %	72.39	72.62	72.53	72.49	0.07	0.695
Breast weight, g	795.50	848.42	854.83	880.50	17.34	0.375
Breast yield, %	28.84	28.39	29.08	29.32	0.25	0.618
Legs weight, g	770.17	826.67	798.00	811.17	13.60	0.521
Legs yield, %	27.93	27.74	27.29	27.00	0.18	0.261

^a^ C = Control (untreated); T1 = in ovo injected with synbiotic (2 mg/embryo); T2 = in ovo injected with synbiotic (3 mg/embryo); T3 = in water synbiotic. SEM = standard error of the mean.

**Table 4 foods-12-02470-t004:** Effect of delivery route and dose of synbiotic on physico-chemical traits of breast muscle of broiler chickens.

	Group ^a^					
	C	T1	T2	T3	SEM	*p*-Value
pH_24_	5.81	5.75	5.74	5.87	0.03	0.367
Color 24 h						
L *	53.50	54.32	52.84	54.13	0.38	0.525
a *	1.43	2.05	1.45	2.03	0.16	0.367
b *	6.93	5.93	5.46	4.63	0.30	0.121

^a^ C = Control (untreated); T1 = in ovo injected with synbiotic (2 mg/embryo); T2 = in ovo injected with synbiotic (3 mg/embryo); T3 = in water synbiotic. SEM = standard error of the mean.

**Table 5 foods-12-02470-t005:** Effect of delivery route and dose of synbiotic on total lipid, cholesterol and fatty acid composition of breast muscle of broiler chickens.

	Group ^a^					
	C	T1	T2	T3	SEM	*p*-Value
Cholesterol, mg/100 g	63.84	61.96	61.86	59.93	1.12	0.715
Lipids, %	1.83	1.52	1.68	1.63	0.07	0.492
Fatty acids ^b^, % of total fatty acids						
C14:0	0.45 ^ab^	0.47 ^ab^	0.55 ^a^	0.40 ^b^	0.02	0.009
C16:0	24.38	23.75	25.68	25.14	0.26	0.041
C16:1	2.41 ^ab^	1.90 ^Bb^	3.09 ^Aa^	2.17 ^b^	0.12	0.001
C17:0	0.04	0.04	0.03	0.04	0.00	0.098
C17:1	0.06	0.06	0.05	0.05	0.00	0.697
C18:0	9.41 ^ab^	10.29 ^a^	8.88 ^ab^	8.27 ^b^	0.20	0.002
C18:1 n-9	31.04	30.72	31.14	33.03	0.39	0.137
C18:2 n-6	24.46	24.13	23.31	24.01	0.22	0.313
C18:3 n-3	1.08	1.13	1.09	1.09	0.02	0.863
C18:3 n-6	0.06	0.05	0.05	0.06	0.00	0.278
C20:0	0.02	0.02	0.02	0.02	0.00	0.736
C20:2 n-6	0.34	0.42	0.33	0.30	0.02	0.043
C20:3 n-6	0.32	0.39	0.28	0.33	0.02	0.116
C20:3 n-3	0.02	0.02	0.01	0.01	0.00	0.039
C20:4 n-6	4.26	4.64	4.01	3.56	0.20	0.281
C20:5 n-3	0.10	0.10	0.11	0.12	0.00	0.604
C22:4 n-6	0.88	1.07	0.80	0.74	0.05	0.065
C22:5 n-3	0.44	0.51	0.38	0.46	0.02	0.295
C22:6 n-3	0.24	0.26	0.19	0.20	0.01	0.204
ƩSFA	34.30	34.59	35.16	33.87	0.28	0.425
ƩMUFA	33.50	32.68	34.29	35.25	0.45	0.219
ƩPUFA	32.20	32.73	30.55	30.88	0.37	0.113
Ʃn-6 PUFA	30.32	30.70	28.76	29.00	0.35	0.126
Ʃn-3 PUFA	1.88	2.03	1.79	1.88	0.03	0.079
Nutritional indices ^c^						
n-6/n-3	16.38	15.20	16.21	15.48	0.22	0.162
PUFA/SFA	0.94	0.95	0.87	0.92	0.01	0.204
AI	0.40	0.39	0.43	0.41	0.01	0.106
TI	0.91	0.91	0.95	0.90	0.01	0.338

^a^ C = Control (untreated); T1 = in ovo injected with synbiotic (2mg/embryo); T2 = in ovo injected with synbiotic (3mg/embryo); T3 = in water synbiotic. ^b^ SFA = saturated fatty acids; MUFA = monounsaturated fatty acids; PUFA = polyunsaturated fatty acids. ^c^ AI = atherogenic index; TI = thrombogenic index. SEM = standard error of the mean. ^A,B^ Means within a row lacking a common superscript differ (*p* < 0.01); ^a,b^ Means within a row lacking a common superscript differ (*p* < 0.05).

## Data Availability

Data presented in this study are available on request from the corresponding author.

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
