# Peer review of "Influence of a Commercial Synbiotic Administered In Ovo and In-Water on Broiler Chicken Performance and Meat Quality"

_foods, 2023, doi:10.3390/foods12132470_

Round 1
Reviewer 1 Report
To whom it may concern,
The current manuscript on "the Influence of a commercial synbiotic administered in-ovo or in-water on broiler chicken performance and meat quality" is about the interesting topic of introducing alternatives to antibiotic use in the poultry diet. However, it needs to be improved based on the comments given in the pdf file. Furthermore, the result of similarity index by turnitin is 44% which needs to be reduced to 20%. Ity can be considered after changes and improvements.
Thank you

Minor improvements are needed.
Reviewer 2 Report
Dear Authors,
I appreciate the opportunity to review your manuscript. Your work contributes to our understanding of the influence of a commercial synbiotic on broiler chicken performance and meat quality. I appreciate the clarity of your introduction and the overall design of your study. However, I have several concerns that need to be addressed for further clarity and consistency in your manuscript.
There is inconsistency in the presentation of the treatment names, C, T1, T2, and S in the abstract (lines 18-19) and C, T1, T2, and T3 in the tables. Please address this.
The "Results and Discussion" section needs to be restructured. In the "Hatchability" subsection, the explanation of the role of synbiotics comes before the presentation of results (line 184). Please first present the results and then discuss them.
Please explain why heavy eggs were selected for injection. The shell quality of heavy eggs is generally low and this might influence hatchability.
The hatch rate in the control group seems unusually high, considering the age of the broiler breeders. Please discuss this inconsistency.
Please provide the average weight of the eggs used in this study.
Please provide a troubleshooting report showing how and at what age the embryos died to account for the 15% difference in hatch rate.
All abbreviations should be presented when they first appear in the text. Please check carefully and correct instances where this was not done.
Please revise line 310 from "Li et al al." to "Li et al."
Please explain your choice to provide more expanded soybean in the finisher diet compared to the grower and starter diets.
Please correct the spelling and capitalization of "coline", "Lysyne", and "energy" in the diet table.
Please present the exact p-value instead of "P>0.05" for tendencies in line 205.
In Table 2, please change "0d" to "d 0" to match the style in other sections.
For clarity, consider adding a note in the tables, such as: "Different superscripts (a, b, c) within the same row indicate significant differences (p < 0.05)."
For completeness, please add the exact p-values to all tables, rather than using "ns" or "*".
I believe addressing these comments will significantly improve the clarity and consistency of your manuscript. I look forward to seeing the revised version of your work.
English quality seems good.
Reviewer 3 Report
Why did the authors choose ‘Scheffe’s test’?
According to the SEM values of the variables from Tables 2 to 5, more statistically significant variables would have been detected if Duncan´s or Bonferroni’s test were considered, thus allowing further description and discussion (comparison) of the variable values.
The authors have published plenty of papers about the theme. The comparison between statistical tests could bring an added value to the theme.
Table 5. Letter assignment for C16:0, C20:2n-6 and C20:3n-3? What about C20:1n-9 (which is 0.40-0.60%), C22:1n-9 and C22:1n-11?
Moderate review needed
Round 2
Reviewer 2 Report
The manuscript is well-written overall. However, I would like to encourage the authors to address the following points:
The statement regarding the weight of chickens from 60 g eggs being 50 g is difficult to believe. It would be helpful if the authors could provide further explanation or clarification on this matter.
On line 178, it is unclear where the hatchability results are presented. I recommend including a table or figure to clearly display this information.
Please ensure consistency in the naming or indicators for the treatments throughout the Table, abstract, and main text. For example, consider using either C T1 T2 T3 or C T1 T2 S consistently.
